# MaSMG7-Mediated Degradation of MaERF12 Facilitates *Fusarium oxysporum* f. sp. *cubense* Tropical Race 4 Infection in *Musa acuminata*

**DOI:** 10.3390/ijms25063420

**Published:** 2024-03-18

**Authors:** Huoqing Huang, Siwen Liu, Yile Huo, Yuzhen Tian, Yushan Liu, Ganjun Yi, Chunyu Li

**Affiliations:** 1Institute of Fruit Tree Research, Guangdong Academy of Agricultural Sciences, Key Laboratory of South Subtropical Fruit Biology and Genetic Research Utilization, Ministry of Agriculture and Rural Affairs, Guangdong Provincial Key Laboratory of Tropical and Subtropical Fruit Tree Research, Guangzhou 510640, China; hqhuang07@163.com (H.H.); liusiwen@gdaas.cn (S.L.); lele940910@163.com (Y.H.); yushanliu2020@outlook.com (Y.L.); 2Guangdong Laboratory for Lingnan Modern Agriculture, Guangzhou 510640, China; 3Maoming Branch, Guangdong Laboratory for Lingnan Modern Agriculture, Maoming 525000, China; 4Guangdong Provincial Key Laboratory of Laser Life Science, College of Biophotonics, South China Normal University, Guangzhou 510631, China; tianyzh@163.com

**Keywords:** banana wilt, *Fusarium oxysporum* f. sp. *cubense* tropical race 4, MaERF12, MaSMG7, cell death

## Abstract

Modern plant breeding relies heavily on the deployment of susceptibility and resistance genes to defend crops against diseases. The expression of these genes is usually regulated by transcription factors including members of the AP2/ERF family. While these factors are a vital component of the plant immune response, little is known of their specific roles in defense against *Fusarium oxysporum* f. sp. *cubense* tropical race 4 (*Foc* TR4) in banana plants. In this study, we discovered that MaERF12, a pathogen-induced ERF in bananas, acts as a resistance gene against *Foc* TR4. The yeast two-hybrid assays and protein-protein docking analyses verified the interaction between this gene and MaSMG7, which plays a role in nonsense-mediated RNA decay. The transient expression of MaERF12 in *Nicotiana benthamiana* was found to induce strong cell death, which could be inhibited by MaSMG7 during co-expression. Furthermore, the immunoblot analyses have revealed the potential degradation of MaERF12 by MaSMG7 through the 26S proteasome pathway. These findings demonstrate that MaSMG7 acts as a susceptibility factor and interferes with MaERF12 to facilitate *Foc* TR4 infection in banana plants. Our study provides novel insights into the biological functions of the MaERF12 as a resistance gene and MaSMG7 as a susceptibility gene in banana plants. Furthermore, the first discovery of interactions between MaERF12 and MaSMG7 could facilitate future research on disease resistance or susceptibility genes for the genetic improvement of bananas.

## 1. Introduction

Banana is a highly valuable tropical and subtropical fruit that enjoys popularity around the world. The crop is especially significant in Africa, where it contributes up to 25–35% of the average daily nutrient intake [1]. It also is considered as the fourth-largest food crop after rice, wheat, and corn by the World Food and Agriculture Organization. While demand for bananas continues to rise, the industry is increasingly threatened by Fusarium wilt, a destructive disease caused by *Fusarium oxysporum* f. sp. *cubense* tropical race 4 (*Foc* TR4). *Foc* TR4 has a devastating economic impact on the banana industry, and due to a lack of resistant resources, monoculture plantations, and global trading, the disease has rapidly spread across the banana-producing countries, such as Mozambique in 2013, Colombia in 2019, and Peru in 2021, which causes great threats to the human diet and nutritional needs. Based on the current alarming rate of spread of *Foc* TR4 in banana production, it is projected to reach 17% of the global banana growing area by 2040, equaling 36 million tons of products worth over USD 10 billion [2]. Currently, farmers rely heavily on fungicides to prevent yield losses caused by fungal diseases, but their extensive use has resulted in the emergence of fungicide-resistant pathogens, raising worries about the residual effects on food, the environment, and human health. Additionally, biological controls have proven unsatisfactory in field experiments [3]. As traditional efforts have failed to effectively control the pathogen, an increased importance has been placed on breeding resistant banana cultivars as a promising method for reducing the destruction caused by banana wilt. Due to the lack of resistant gene resources, banana molecular resistance breeding is difficult. Therefore, more attention should be placed on the identification of resistance-related genes.

Previous studies have uncovered a variety of resistance genes in other plants and demonstrated their potential in fighting diseases. For example, the receptor-like kinase SDS2 was found to positively regulate cell death and immunity in rice, allowing plants to better resist pathogenic attacks [4]. A separate study of rice reported that the phosphorylation of IPA1 induced by the fungi *Magnaporthe oryzae* activates the expression of WRKY45, a pathogen defense gene. This triggers a temporarily enhanced disease resistance, and roughly 48 h post-infection, IPA1 returns to a non-phosphorylated state and resumes its growth-supporting roles [5]. Furthermore, in apples, MdUGT88F1-mediated phloridzin biosynthesis is found to regulate resistance to *Valsa Canker* [6]. In addition to breeding for resistance genes, new genome editing techniques have allowed researchers to create novel disease-resistant plants by manipulating susceptibility genes [7]. For instance, a susceptibility gene known as wheat kinase TaPsIPK1 is targeted by the PsSpg1 effector produced by *Puccinia striiformis* f. sp. *Tritici* to promote infection. Inactivating TaPsIPK1 confers robust rust resistance without any growth or yield penalty [8]. In potatoes, silencing the susceptibility genes *StDND1*-, *StDMR1*-, and *StDMR6* leads to increased late blight resistance [9]. Moreover, maize can be protected against southern leaf blight by knocking out a leucine-rich repeat receptor kinase gene named *ChSK1* (Cochliobolus heterostrophus Susceptibility Kinase 1) [10]. However, there is a poor understanding of the mechanisms by which *Foc* TR4 interacts with banana hosts.

Various plant processes are governed by families of plant-specific transcription factors (TFs), of which APETALA2/ethylene response factor (AP2/ERF) is the largest [11]. This family plays an important role in many development processes, such as embryo formation [12], floral growth [13], and both biotic and abiotic stresses [14,15,16]. A growing body of research has indicated that AP2/ERF family members positively respond to pathogenic infection, making them excellent candidates for use in improving biotic stress resistance in crops. A previous study in wheat found that an ERF gene known as TaPIEP1 is induced upon infection, leading to enhanced resistance against *Bipolaris sorokiniana* through overexpression [17]. Additionally, OsERF101 in rice regulates NLR Xa1-mediated immunity, which is induced by the perception of TAL effectors [18]. AP2/ERFs integrates hormonal signaling to regulate plant disease resistance, such as regulating SA biosynthesis or signaling pathways. For instance, MdERF11 positively regulates defense responses against *Botryosphaeria dothidea* by promoting SA biosynthesis in apples [19]. However, some AP2/ERF family members negatively regulate resistance against plant pathogens. For example, OsERF922 in rice can be rapidly and strongly induced in response to *M*. *oryzae*. A knockout of this gene is found to enhance *M*. *oryzae* resistance by upregulating the expression of defense-related genes, such as PR and PAL. Conversely, the overexpression of OsERF922 leads to a decreased expression of these genes, resulting in increased sensitivity to *M*. *oryzae* [20]. AtERF9 is a transcription repressor binding to the GCC-box of *PATHGEN*-*INDUCIBE PLANT DEFENSIN* (*PDF1.2*) gene, and knocking out AtERF9 significantly promotes *AtPDF1.2* expression, resulting in enhanced resistance to *Botrytis cinerea* [21]. Moreover, some AP2/ERF transcription factors contain a conserved (L/F)DLN(L/F)xP motif, also named the EAR (ERF-associated amphiphilic repression) motif, and inhibit target gene expression [22]. For instance, silencing StERF3 in tomato promotes the expression of defensive genes (PR1, NPR1, and WRKY1), resulting in enhanced immunity to *Phytophthora infestans* [23]. Although these findings collectively suggest an important link between AP2/ERFs and pathogenic infection, few disease-related ERF genes have been reported in banana plants.

In this study, MaERF12 is a pathogen-induced gene contained an EAR motif. In order to study the roles of MaERF12 between the interactions of *Foc* TR4 and bananas, we assessed how the MaERF12 resistance gene and the MaSMG7 susceptibility gene respond to *Foc* TR4 infection in banana plants. Furthermore, we also elucidated the functions of both genes by examining their physical interactions, measuring their impact on immune responses in *Nicotiana*. *benthamiana* leaves, and determining the effects of their gene silencing in banana leaves during *Foc* TR4 infection. Our results verify the functions of MaERF12 and MaSMG7 in response to a pathogenic invasion, providing valuable new insights into the molecular mechanisms underlying plant immune defense. This study also provides more targets of molecular disease-resistance breeding for the future genetic improvement of bananas.

## 2. Results

### 2.1. MaERF12 Positively Regulates Foc TR4 Resistance in Banana

In this study, we identified 315 AP2/ERF family members associated with the response of *M*. *acuminata* to *Foc* TR4 infection (Appendix A). Among them, a pathogen-induced gene MaAP2-210, also named MaERF12 below, displayed notably higher expression in the resistant cultivar compared to the susceptible one (Appendix A; Figure 1a). Although the gene is induced during infection, it is unclear if it actively participates in the immune response. To investigate this possibility, a transient expression assay was conducted in *N*. *benthamiana* leaves. The findings reveal that MaERF12 had GFP signals in the nucleus colocalized with the nuclear marker NLS-mcherry (Figure 1b). It is capable of inducing strong cell death in *N. benthamiana* leaves, resembling the effects of the *Phytophthora infestans* elicitor INF1 (Figure 1c). The DAB staining results show that MaERF12 could induce strong ROS accumulation (Figure 1d). An additional assay was performed to determine whether MaERF12-induced cell death relies on the receptor-like kinases BAK1 and SOBIR1, which are vital components of defense signaling [24,25]. The results demonstrate that both MaERF12 and INF1 can induce cell death in wild-type *N*. *benthamiana* leaves; however, only MaERF12 can induce strong cell death in the mutant plants (Figure 1e,f). This suggests that MaERF12-induced cell death does not rely on SOBIR1 or BAK1 pathways. Due to the challenges associated with genetically transforming banana plants, the biological functions of MaERF12 were investigated during *Foc* TR4 infection by transiently silencing the genes. *MaERF12*-dsRNAs were topically applied to banana leaves to determine whether the gene positively or negatively affects disease resistance. The RT-qPCR analysis showed that the expression level of *MaERF12* in dsRNA-treated leaves was higher than in water-treated leaves (Figure 1g). The dsRNA-treated leaves exhibited larger patches of necrosis compared to the water controls (Figure 1h–j), indicating that *MaERF12* gene silencing could reduce *Foc* TR4 resistance in banana plants. Therefore, we concluded that MaERF12 positively regulates disease resistance against *Fusarium oxysporum* f. sp. *cubense* tropical race 4.

### 2.2. MaSMG7 Degrades MaERF12 and Negatively Regulates Banana Disease Resistance to Foc TR4

To investigate the molecular mechanisms of MaERF12 in bananas, we conducted a series of yeast two-hybrid assays. Our initial Y2H screenings identified MaSMG7, a nonsense-mediated mRNA decay factor that restricts cell death induction during pathogen infection [26], as a candidate interactor of MaERF12. The relationship between the two proteins was verified with further Y2H assays (Figure 2a). Furthermore, the protein-protein docking analysis demonstrated that MaERF12 interacted with MaSMG7 with a high confidence score of 0.9572 (Figure 2b). The subcellular location assays demonstrated that MaSMG7 might localize along with MaERF12 in p-bodies, but not in the nucleus (Figure 2c,d). Interestingly, in contrast to the expression pattern of MaERF12, MaSMG7 was induced in both resistant and susceptible cultivars, but more highly so in the latter (Figure 2e). Therefore, we speculated that the function of MaSMG7 might be antagonistic to that of MaERF12. To test this hypothesis, MaSMG7-dsRNAs were applied to banana leaves, successfully silencing the gene analyzed by RT-qPCR (Figure 2f). Less necrosis was observed in MaSMG7-dsRNA-treated leaves compared to water-treated leaves (Figure 2g–i), demonstrating that silencing MaSMG7 significantly improved resistance following *Foc* TR4 inoculation.

Next, MaSMG7 and MaERF12 were expressed both independently and together in *N*. *benthamiana* leaves to explore their interaction mechanisms. The findings show that MaSMG7 alone could not induce cell death (Figure 2j) but was able to inhibit INF1- and MaERF12-induced cell death (Figure 2k,l). This suggests that MaSMG7 might play a role in inhibiting plant immunity. Furthermore, immunoblot analyses revealed that the GFP:MaERF12 protein level was significantly lower in tissues co-expressing MaSMG7:FLAG compared to the control co-expressing the FLAG empty vector (Figure 2m). Pretreating leaves with MG132 did not trigger similar changes in GFP:MaERF12 protein levels (Figure 2m), suggesting that MaERF12 degradation might be effectively inhibited by MG132 through the 26S proteasome pathway. Additionally, the protein-nucleic acid docking showed that the MaSMG7 protein bound to *MaERF12* mRNA with a high confidence score of 0.9649 (Figure 2n). Therefore, in bananas, MaSMG7 was determined to be a susceptibility gene for *Foc* TR4 infection by affecting MaERF12 accumulation.

## 3. Discussion

Throughout the plant kingdom, various AP2/ERF family members are known to regulate immune responses during pathogenic infection. Notable examples include TaPIEP1 in wheat against *Bipolaris sorokiniana* [17], OsERF101 in rice against *Xanthomonas oryzae* pv *oryzae* [18], and ZmERF105 in maize against *Exserohilum turcicum* [27]. However, no such genes conferring resistance against *Foc* TR4 have been reported in bananas. In this study, we confirm the resistance properties of the AP2/ERF transcription factor MaERF12 and the susceptibility properties of the nonsense-mediated mRNA decay factor MaSMG7. Both genes were found to influence the pathogenicity of Fusarium wilt in banana and were significantly induced during *Foc* TR4 infection. Furthermore, MaSMG7 was found to interact with and degrade MaERF12. These results will deepen our understanding of molecular mechanisms of *Foc* TR4–banana interactions, and provide more gene resources for disease resistance breeding.

*Foc* TR4 is a hemi-biotrophic pathogen that strategically delivers a series of effectors to manipulate host immunity and facilitate successful colonization [28,29]. Upon infection, plants often induce rapid cell death to limit the spread of pathogens. This method of disease resistance is employed to defend against biotrophic or hemi-biotrophic pathogens [30], but is not effective against necrotrophic pathogens [26,31,32]. In this study, the inoculation of banana with *Foc* TR4 resulted in a significant induction of MaERF12, and the gene expression in *N*. *benthamiana* led to significant cell death. This suggests that MaERF12 plays a positive regulatory role in banana immune responses. Furthermore, the silencing of the MaERF12 gene was found to enhance susceptibility in banana leaves, indicating that the gene has potential for use in crop genetic improvement. Conversely, proteins inhibiting cell death have been reported to improve plant disease resistance against necrotrophic fungi, but it is unknown how these functions differ in hemi-biotrophic fungi during different infection stages. It is hypothesized that these proteins may negatively regulate disease resistance during the early stages of infection and positively regulate it in the later stages. In this study, MaSMG7 was found to be induced in the early stage of *Foc* TR4 infection, leading to the inhibition of immune responses such as INF1- and MaERF12-induced cell death. Taken together, these results suggest that MaSMG7 functions as a susceptibility gene, and that silencing its expression could enhance Fusarium wilt resistance in banana plants. Previous research has demonstrated that plant pathogens may hijack susceptibility genes to disrupt regulation and suppress the host immune response. The identification of these genes holds immense value, as they can be manipulated with new genome editing techniques to improve disease resistance in crops without significantly affecting the desired agronomic traits [8,33]. Therefore, *MaSMG7* could be edited in future studies on generating resistant banana plants.

In *Arabidopsis thaliana*, SMG7 influences the initiation of nonsense-mediated mRNA decay by providing a connection to mRNA degradation machinery and serving as an adapter for UPF protein function [34,35,36]. Further research is required to gain a stronger understanding of whether MaSMG7 is involved in the nonsense-mediated mRNA decay of MaERF12, and how it inhibits MaERF12-induced cell death. A growing body of research indicates that single-nucleotide polymorphisms (SNPs) have a wide-ranging impact on various life processes, including disease resistance [37,38,39]. It is theorized that SNPs may not only influence the binding abilities between MaERF12 mRNA and MaSMG7 proteins, but they may also mutate amino acids in both proteins in resistant and susceptible cultivars. These mutations could lead to variations in protein–protein binding abilities, potentially causing MaSMG7-mediated degradation, and resulting in cultivar-specific differences in MaERF12 accumulation. These results suggest that the MaSMG7-driven degradation of MaERF12 may explain its low expression levels in susceptible banana cultivars. The opposite effect was observed in resistant cultivars, further supporting this hypothesis.

In summary, we confirmed MaERF12 as a resistance gene and MaSMG7 as a susceptibility gene in banana plants, suggesting potential for gene modification using banana genetic improvement techniques. The interactions between these genes were characterized, and MaSMG7 was found to degrade MaERF12 through the 26S proteasome pathway. These findings highlight the potential of using MaERF12 and MaSMG7 as targets for developing resistant banana germplasms, ultimately reducing crop loss caused by *Foc* TR4 and increasing the economic value of the banana industry. Additionally, gene inhibitors will be developed to control Fusarium wilt in bananas based on the understanding of the molecular function of MaERF12 and MaSMG7.

## 4. Materials and Methods

### 4.1. Plant and Fungi Materials, Constructs, and Bacterial Strains

The Cavendish banana (*Musa* spp. AAA group) cultivars “Brazilian” (BX) and ”ZhongJiao No.6” (ZJ6) were utilized as resistant and susceptible materials, respectively. ZJ6 was artificially bred from BX and was obtained from the Fruit Tree Research Institute, Guangdong Academy of Agricultural Sciences, China. Plants were grown to the 5 to 6-leaf stage in a greenhouse maintained at 28 °C under a 14 h light/10 h dark cycle. The *N*. *benthamiana* plants were grown in a greenhouse maintained at 24 °C for roughly 28 d under a 16 h light/8 h dark cycle. *Foc* TR4 strain II5 (NRRL#54006) with a clear genetic background was obtained and cultured either in potato dextrose agar (PDA) for 6 d or in potato dextrose broth (PDB) for 3 d. The *Agrobacterium tumefaciens* strain GV3101 (Weidi Biotech Co., Ltd., Shanghai, China) was used for transient expression in *N*. *benthamiana* leaves. GV3101 strains were cultured overnight at 28 °C in liquid Luria-Bertani (LB) medium supplemented with 50 mg.L^−1^ rifampicin and 50 mg.L^−1^ kanamycin.

### 4.2. Identification of AP2/ERF Gene Family

*M. acuminata* amino acids were downloaded from the Banana Genome Database (http://banana-genome-hub.southgreen.fr/, accessed on 22 September 2023). A Hidden Markov Model (HMM) for ERF (PF00847) was downloaded from the Pfam database (http://pfam.xfam.org/, accessed on 22 September 2023). The *M*. *acuminata* genome was then searched using PF00847 as a query with the Simple HMM Search in TBtools [40]. The NCBI CDD search (https://www.ncbi.nlm.nih.gov/cdd, accessed on 7 October 2023) was used for domain validation in the resulting sequences. TBtools was used to construct the phylogenetic analysis.

### 4.3. Plasmid Construction

Genes of interest were cloned from cDNA extracted from BX roots using 2 × Phanta Max Master Mix according to the manufacturer’s instructions (Vazyme Biotech Co., Ltd., Nanjing, China). Amplified fragments were ligated into the BamHI-digested pCAMBIA1300-35S:GFP empty vector, XbaI-digested pCAMBIA1300-35S:CFP, SacI and HindIII-double digested pCambia35S:4Myc:3FLAG, or the SmaI-digested pGBKT7 or pGADT7 empty vector using In-Fusion Cloning Kits (Vazyme Biotech Co., Ltd., Nanjing, China). Individual colonies containing each construct were verified via PCR and sequencing (Sangon Biotech Co., Ltd., Shanghai, China). *Phytophthora infestans* elicitor INF1 (XM_002900382.1) was artificially synthesized by Sangon Biotechnology (Shanghai) Company and subsequently cloned into the BamHI-digested pCAMBIA1300-35S:GFP empty vector. All primers used to generate constructs are listed in Appendix A.

### 4.4. RNA Extraction and RT-qPCR

Total RNA was extracted from banana roots using RNA out kits (Accurate Biotech Co., Ltd., Changsha, China) according to the manufacturer’s instructions. Reverse transcription was performed using the Evo M-MLV One-Step RT-PCR Kit (Accurate Biotech Co., Ltd., Changsha, China). RT-qPCR was performed in a StepOne RT-PCR system (Applied Biosystems, Waltham, MA, USA) with the ChamQ Universal SYBR qPCR Master Mix (Vazyme Biotech Co., Ltd., Nanjing, China). *MaTUB* (β-tubulin) was employed as the endogenous banana reference gene. The expression levels of each gene were determined using the 2^−∆∆CT^ method. The expressions of all samples were examined in three biological replicates with three technical repetitions. All primer pairs are listed in Appendix A.

### 4.5. Transcriptomic Analysis of Banana Genes

RNA-seq analysis was performed with BX and ZJ6 root samples following 18 h, 32 h, and 56 h of *Foc* TR4 II5 inoculation. Three replicates were generated for each cultivar at each time point. Illumina HiSeq × 10 was used to generate 150 bp paired-end reads and sequencing was performed by Gene Denovo Biotechnology Company (Guangzhou, China). Filtered reads were aligned to the *M*. *acuminata* genome using Hisat2 v.2.0.5 [41]. Differential expression analyses between BX and ZJ6 were conducted using the DESeq2 R package. Transcripts with a Log_2_FoldChange greater than 1 or less than −1 (*p* < 0.05) were considered to be differentially expressed.

### 4.6. Transient Gene Expression in N. Benthamiana

Agrobacterium-mediated transient expression was carried out in *N*. *benthamiana* leaves using a modified method as previously described [42]. The generated constructs were transformed independently into *A*. *tumefaciens* GV3101 and cultured for 8 h in liquid LB medium at 28 °C. Bacterial cells were harvested through centrifugation and then re-suspended in an infiltration solution containing 10 mM 2-(4-Morpholino) ethanesulfonic acid, 10 mM MgCl_2_, and 200 μM acetosyringone (pH 5.6). The OD_600_ of each strain was adjusted to 0.6 and incubated at room temperature for 2–3 h before infiltration into leaves of 4-week-old *N*. *benthamiana* plants. During the transient expression assay, necrosis was observed in the leaves of *sobir1* [43] and *bak1* [44,45] knockout *N*. *benthamiana* mutants as well as a wild-type control for 3 days post-infiltration. Subcellular location assays in wild-type or NLS-mcherry transgenic *N*. *benthamiana* were performed for 3 days post-infiltration using a confocal microscope (LSM 710, Carl Zeiss, Oberkochen, Germany). Western blot analyses were conducted 72 h post-infiltration. All samples were examined in three biological replicates.

### 4.7. dsRNA-Mediated Gene Silencing in Banana

dsRNAs of interest complementary to the selected genes were synthesized using a T7 RNAi Transcription Kit (Vazyme Biotech Co., Ltd., Nanjing, China) according to the manufacturer’s instructions. PCR was employed to introduce the T7 promoter sequence at both the 5′ and the 3′ ends of the RNAi target fragments. The resulting fragments were purified and used as templates for dsRNA synthesis. Infection assays were performed on detached banana leaves to determine whether the application of dsRNAs to plant surfaces could confer effective protection against *Foc* TR4 [46]. Briefly, the 5 to 6-leaf stage banana leaves were cross cut with a blade and treated with a mixture of 10 μL dsRNA (500 ng.μL^−1^) and 0.02% Silwet L-77. Approximately 1 h after treatment, the leaves were inoculated with 5 mm-diameter mycelial plugs obtained from the growing edge of a 6-day-old *Foc* TR4 II5 strain culture. The side of the mycelium touched the banana leaves. The dsRNA and water were applied on the same leaf. Ten leaves were used in each experiment. Subsequently, the leaves were placed in a plastic tray on top of two layers of wet paper napkins and covered with a plastic film to maintain high humidity. The trays were stored at 28 °C in the dark. The leaves were photographed at 10 d post-inoculation (dpi) with a Nikon camera. Fungal lesions were measured using ImageJ v1.8.0 software. All primer pairs are listed in Appendix A.

### 4.8. Trypan Blue Staining

Trypan blue staining was performed according to a modified method based on Fernández-Bautista et al. [47]. Banana leaves with II5 infection were soaked at room temperature overnight in a trypan blue solution (0.02% trypan blue in a volumetric ratio of 1:1:1:1:8 phenol:glycerol:lactic acid:water:ethanol). Samples were destained repeatedly in 75% alcohol until the solution achieved a consistently clear color. Samples were then photographed with a Nikon camera.

### 4.9. Yeast Two-Hybrid Assays

Yeast two-hybrid (Y2H) assays were conducted to identify protein-protein interactions in vitro. The CDS of MaERF12 and MaSMG7 were cloned in-frame into the bait vector pGBKT7 and the prey vector pGADT7, respectively, using In-Fusion Cloning Kits (Vazyme Biotech Co., Ltd., Nanjing, China). The resulting bait and prey vectors were co-transformed into the Y2HGold strain (WeidiBio Biotech Co., Ltd., Shanghai, China) to identify positive interactions, according to the manufacturer’s protocol (Clontech, Mountain View, CA, USA). The pGBKT7-53 + pGADT7-T and pGBKT7-lam + pGADT7-T co-transformants were employed as the positive and negative controls, respectively. Positive interactors were screened using an SD/–leucine–tryptophan–adenine–histidine medium and positive transformants were screened using an SD/–leucine–tryptophan medium. X-α-gal (40 mg.L^−1^) was used to confirm positive interactions by turning colonies blue. Plates were incubated at 28 °C in the dark for 3 days. Plates were then photographed using a Nikon camera. All primers used for plasmid constructions are listed in Appendix A.

### 4.10. Molecular Docking

The interactions of two molecules were predicted by the HDOCK SERVER (http://hdock.phys.hust.edu.cn/, accessed on 25 January 2024). The amino acids of MaERF12 were uploaded as a ligand molecule to the website, while MaSMG7 was uploaded as a receptor molecule. The model with the highest confidence score was used for image production with PyMol v2.5.7 software (https://pymol.org/2/, accessed on 25 January 2024).

## Figures and Tables

**Figure 1 ijms-25-03420-f001:**
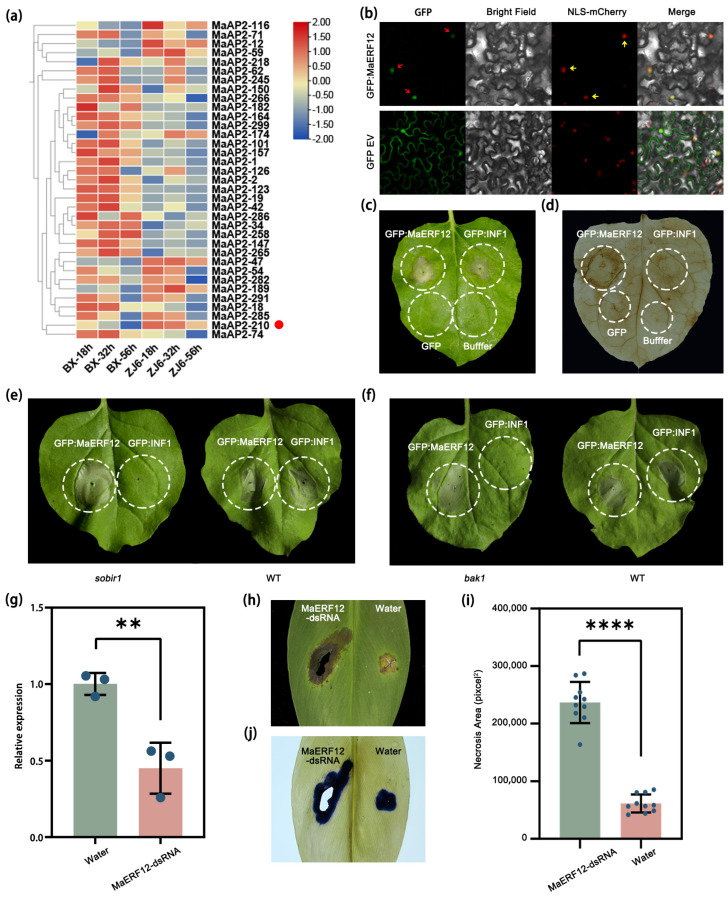
Regulatory effects of MaERF12 on the banana immune response. (**a**) Pathogen-induced AP2/ERF genes in banana roots. Heatmap of AP2/ERF genes in BX and ZJ6 banana roots inoculated with II5 strains for 18 h, 32 h, and 56 h. (**b**) Subcellular localization in NLS-mCherry transgenic *N*. *benthamiana* leaves. Photographs taken at 3 d post agro-infiltration. The GFP empty vector was used as a control. Scale bar = 20 μm. (**c**) Necrosis symptoms observed in *N*. *benthamiana* leaves 3 d post agro-infiltration. (**d**) DAB staining of *N*. *benthamiana* leaves. (**e**,**f**) Necrosis symptoms of MaERF12 and INF1 agro-infiltration in wild-type, (**e**) *sobir1*, and (**f**) *bak1 N*. *benthamiana* leaves 3 d post agro-infiltration. (**g**) Expression level of MaERF12 in dsRNA- and water-treated banana leaves determined by qRT-PCR. Data are presented as means ± SDs (two-tailed Student’s *t*-test, “**” indicates a significant difference at *p* < 0.01, n = 3). (**h**) Disease symptoms and (**i**) necrosis area of dsRNA- and water-treated banana leaves. Photographs taken 10 d post agro-infiltration. Necrosis area was calculated with ImageJ v1.8.0 software. Data are presented as means ± SDs (two-tailed Student’s *t*-test, “****” indicates a significant difference at *p* < 0.0001, n = 10). (**j**) Trypan blue staining of banana leaves. Blue-stained areas indicate tissue necrosis caused by *Foc* TR4.

**Figure 2 ijms-25-03420-f002:**
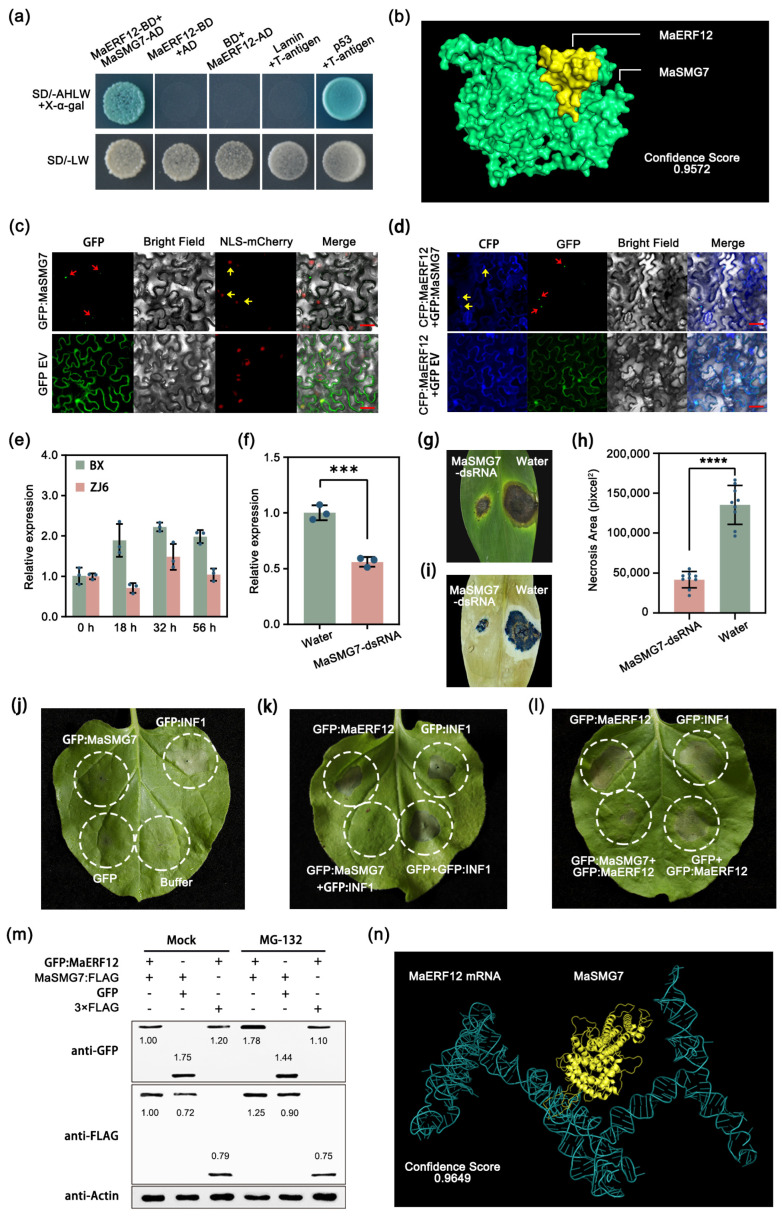
Effects of MaSMG7 on susceptibility and MaERF12 degradation. (**a**) MaERF12 interacts with MaSMG7 identified with Y2H. SD/-LWAH represents SD/-leucine-tryptophan-adenine-histidine and SD/-LW represents SD/-leucine-tryptophan. The SD/-LWAH medium was combined with X-α-gal. (**b**) The protein-protein docking of MaSMG7 and MaERF12. The cyan pattern indicates MaSMG7 protein, the yellow pattern indicates MaERF12 mRNA. (**c**) Subcellular localization in NLS-mCherry transgenic *N*. *benthamiana* leaves. Photographs taken at 3 d post agro-infiltration. GFP empty vector was used as a control. Scale bar = 20 μm. (**d**) Co-localization of GFP:MaSMG7 and CFP:MaERF12 in wild-type *N*. *benthamiana* leaves. Photographs taken 3 d post infiltration. Co-expressions of GFP empty vector (GFP EV) and CFP:MaERF12 were used as a control. Scale bar = 20 μm. (**e**) qRT-PCR analysis showed that MaERF12 was induced during *Foc* TR4 infection. Roots of BX and ZJ6 were sampled at 18 h, 32 h, and 56 h. (**f**) Expression levels of MaSMG7 in dsRNA- and water-treated banana leaves determined by qRT-PCR analysis. Data are presented as means ± SDs (two-tailed Student’s *t*-test, “***” indicates a significant difference at *p* < 0.001, n = 3). (**g**) Disease symptoms and (**h**) necrosis area of dsRNA- and water-treated banana leaves. Photographs taken 10 dpi. Necrosis area was calculated with ImageJ v1.8.0 software. Data are presented as means ± SDs (two-tailed Student’s *t*-test, “****” indicates a significant difference at *p* < 0.0001, n = 10). (**i**) Trypan blue staining of banana leaves. (**j**) *A*. *tumefaciens* leaves infiltrated with GFP empty vector, MaSMG7, and INF1. (**k**) Leaves infiltrated with MaERF12, INF1, co-expression of MaSMG7 and INF1, and co-expression of MaSMG7 and GFP empty vector. (**l**) Leaves infiltrated with MaERF12, INF1, co-expression of MaSMG7 and MaERF12, and co-expression of MaSMG7 and GFP empty vector. (**j**–**l**) Scale bar = 20 μm, photographs taken at 3 d post agro-infiltration. (**m**) Degradation in co-expression of MaSMG7:FLAG and GFP:MaERF12, MaSMG7:FLAG and GFP empty vector, and GFP:MaERF12 and FLAG empty vector. Leaves harvested 2 d post-infiltration. MG132 (50 μM) was infiltrated as a control at 18 h prior to sampling. (**n**) Molecular docking of MaSMG7 protein and *MaERF12* mRNA. Yellow pattern indicates MaSMG7 protein, blue pattern indicates *MaERF12* mRNA.

## Data Availability

Data contained within the article.

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
