# Peer review of "MaSMG7-Mediated Degradation of MaERF12 Facilitates Fusarium oxysporum f. sp. cubense Tropical Race 4 Infection in Musa acuminata"

_ijms, 2024, doi:10.3390/ijms25063420_

Round 1
Reviewer 1 Report
Comments and Suggestions for Authors
In this work, Huang et al. investigated the banana plant's reactions to Foc TR4 infection by studying the MaERF12 resistance gene and the MaSMG7 susceptibility gene. By examining the physical interaction of two genes, evaluating their impact on the immunological responses of Nicotiana benthamiana leaves, and studying the effects of Foc TR4 infection on banana leaves, researchers were able to elucidate the functions of both genes.
The paper begins with a thorough review of the background literature and proceeds to a methodological design that has been meticulously developed and executed. The paper was written very carefully. So, my decision is accepted in its present form.
Reviewer 2 Report
Comments and Suggestions for Authors
The paper, titled "MaSMG7-mediated degradation of MaERF12 facilitates Fusarium oxysporum f. sp. cubense tropical race 4 infection in Musa acuminata", discusses the role of specific genes in banana plants and the interaction between MaSMG7-mediated degradation of MaERF12 facilitates Fusarium oxysporum f. sp. cubense tropical race 4 (Foc TR4) infection in Musa acuminata. The study investigates the interaction between MaERF12, a resistance gene, and MaSMG7, a susceptibility gene, and highlights their influence on the immune response in banana plants.
General comments:
The summary provides a concise overview of the aims, methods and main findings of the study. However, it could be improved by emphasising more clearly the novelty and significance of the research.
The introduction sets the context well and emphasises the importance of the study. However, it could be improved by providing more background information on Fusarium wilt in banana plants and the importance of understanding the molecular mechanisms underlying resistance and susceptibility.
The results section is comprehensive and well organised and effectively presents the experimental results together with their interpretations.
The discussion section effectively discusses the implications of the results and relates them to the existing literature. However, it could benefit from a clearer delineation between the contributions of the study and previous research.
The "Materials and Methods" section contains detailed information on the experimental procedures that are essential for reproducibility. However, it could be organised more clearly to improve readability.
Specific points for improvement:
Abstract:
Emphasise the novelty of the results, particularly in relation to the identification of MaERF12 as a resistance gene and MaSMG7 as a susceptibility gene in banana plants.
Introduction:
Give a brief overview of Fusarium wilt in banana plants, its economic importance and its impact on global banana production.
Explain the objectives of the study in more detail, including the specific objectives related to understanding the role of MaERF12 and MaSMG7 in Fusarium wilt resistance and susceptibility.
Provide more background information on the AP2/ERF family of transcription factors and highlight their importance in plant immune responses and their potential as targets for crop improvement.
Results:
Provide a brief summary or interpretation of the key findings of each figure in the text to aid understanding.
Discussion:
Clearly explain the significance of the results of the study to our understanding of the mechanisms of Fusarium wilt resistance in banana plants.
Explain how the results contribute to existing knowledge in this area and address any gaps or limitations identified during the study.
Consider possible implications of the results for banana breeding programmes and strategies to control Fusarium wilt in agriculture.
Materials and Methods:
Explain any changes or adaptations made to standard protocols and provide references where appropriate.
Include information on sample size, replicates and statistical analyses for each experiment.
